# Diagnostic Performance of ^99m^Tc-iPSMA SPECT/CT in the Initial Staging of Patients with Unfavorable Intermediate-, High-, and Very High-Risk Prostate Cancer: A Comparative Analysis with ^18^F-PSMA-1007 PET/CT

**DOI:** 10.3390/cancers15245824

**Published:** 2023-12-13

**Authors:** Joel E. Vargas-Ahumada, Sofía Denisse González-Rueda, Fabio Andrés Sinisterra-Solís, Pamela Casanova-Triviño, Quetzali Pitalúa-Cortés, Irma Soldevilla-Gallardo, Anna Scavuzzo, Miguel Angel Jimenez-Ríos, Francisco Osvaldo García-Pérez

**Affiliations:** 1Nuclear Medicine Department, Instituto Nacional de Cancerología, Tlalpan, Mexico City 14080, Mexico; drjoelvargas7@gmail.com (J.E.V.-A.); denissegr205@gmail.com (S.D.G.-R.); fabiosinisterra128@gmail.com (F.A.S.-S.); zafily3@gmail.com (P.C.-T.); quecho70@hotmail.com (Q.P.-C.);; 2Urologic Sugery Department, Instituto Nacional de Cancerología, Tlalpan, Mexico City 14080, Mexico; annasc80@gmail.com (A.S.); incanurologia@gmail.com (M.A.J.-R.)

**Keywords:** prostate cancer, PSMA, SPECT/CT

## Abstract

**Simple Summary:**

The main objective of this study was to compare the diagnostic accuracy of ^99m^Tc-iPSMA SPECT/CT with that of ^18^F-PSMA-1007 PET/CT in patients with unfavorable intermediate- and high-risk prostate cancer for primary diagnoses and its impact on clinical staging. Our study showed that ^99m^Tc-iPSMA SPECT/CT is useful in the primary diagnosis of PCa. Despite its slightly lower lesion detectability compared to ^18^F-PSMA, it did not impact the clinical staging and, consequently, the initial treatment intention. Our results, if validated by further studies, may encourage the use of ^99m^Tc-iPSMA as a more economical and available alternative for staging prostate cancer.

**Abstract:**

Prostate cancer is a leading cause of cancer death in men worldwide. Imaging plays a key role in disease detection and initial staging. Emerging data has shown the superiority of PSMA imaging with PET/CT over conventional imaging for primary diagnoses. Single photon emission computed tomography is more available worldwide, and the imaging agent is low in cost. The aim of this study is to compare the diagnostic accuracy of ^99m^Tc-EDDA/HYNIC-iPSMA SPECT/CT to ^18^F-PSMA-1007 PET/CT in the primary diagnosis of prostate cancer and the impact on clinical staging. Methods: In this prospective controlled study, 18 patients with histologically confirmed prostate cancer with unfavorable intermediate-, high-, and very high-risk characteristics were recruited to undergo ^18^F-PSMA-PET/CT and ^99m^Tc-iPSMA SPECT/CT. The median age of the patients was 71 years old, and the median PSA level was 23.3 ng/mL. Lesions were divided into the prostate, seminal vesicles, lymph nodes, bone, and visceral metastases. Volumetric analysis was also performed between the two imaging modalities and correlated with PSA levels. Results: A total of 257 lesions were detected on ^18^F-PSMA-PET/CT: prostate (n = 18), seminal vesicles (n = 12), locoregional lymph nodes (n = 62), non-locoregional (n = 67), bone (n = 90), and visceral (n = 8). Of these, ^99m^Tc-iPSMA-SPECT/CT detected 229 lesions, while both reviewers detected 100% of the lesions in the prostate (18/18), seminal vesicles (12/12), and visceral (8/8); LN LR (56/62; 90%), NLR (57/67; 85%), and bone (78/90; 86%). There were no statistically significant differences between volumetric parameters (*t* = −0.02122; *p* = 0.491596). Conclusions: ^99m^Tc-iPSMA SPECT/CT is useful in the primary diagnosis of prostate cancer. Despite it showing a slightly lower lesion detection rate compared to ^18^F-PSMA PET/CT, it exhibited no impact on clinical staging and, consequently, the initial treatment intention.

## 1. Introduction

Prostate cancer (PCa) is a leading cause of cancer and is the second most common cancer diagnosis made in men worldwide. The prevalence and incidence of this neoplasm have increased over the years; an estimated 288,300 new cases of prostate cancer will be diagnosed in the United States in 2023 and will correspond to 5.7% of all cancer deaths [1,2]. Early detection of primary disease is highly relevant in terms of prognosis and therapy management [3]. Multiple treatment options are available for patients with PCa according to their risk of disease, which is usually determined by Gleason score, PSA level, and rectal examination [4]. Based on the American Urological Association (AUA) and the European Association of Urology (EAU) guidelines, unfavorable intermediate-risk is defined as T2b-c, ISUP 2-3, a total prostate-specific antigen (tPSA) between 10 and 20 ng/mL, and more than 50% of positive biopsy cores. Alternatively, high- and very high-risk are defined as having T3-T4, tPSA >20 ng/mL and/or a histological grade according to the International Society of Urological Pathology (ISUP) of 4/5 [5]. Previous studies have found that predicting distant metastases based on tPSA is not reliable [6], and the probability of mortality in high-risk patients does not always correlate with biochemical parameters. Therefore, imaging studies for the initial staging evaluation are mandatory. NCCN guidelines also recommend imaging studies for symptomatic patients and/or those with a life expectancy greater than five years in patients with an unfavorable intermediate risk [7].

The overexpression of prostate-specific membrane antigen (PSMA) occurs in PCa, and its expression has been correlated with the malignancy, aggressiveness, and DNA repair pathways of these tumors [8,9,10]. This glycoprotein has been targeted using a variety of inhibitors and small molecules, most commonly labeled with ^68^Ga and ^18^F. During the last decade, PSMA-targeted positron emission tomography (PET) has become a substantial part of the imaging of prostate cancer. It has demonstrated higher accuracy for disease localization in individuals with newly diagnosed PCa compared with conventional imaging (bone scan and computed tomography), showing 27% greater accuracy [11]. Also, PSMA PET/CT showed high specificity (≥95%) for the detection of pelvic lymph node metastases in individuals undergoing radical prostatectomy [12,13]. This imaging modality has also demonstrated metastatic lesions in 55% of patients previously diagnosed with non-metastatic castration-resistant prostate cancer by conventional imaging [14].

Despite the success of this imaging modality, there are challenges in meeting the high demand and expensive costs, especially in low- and middle-income countries. These challenges have awakened interest in ^99m^Tc-labeled PSMA inhibitors. Single photon emission computed tomography fused with computer tomography (SPECT/CT) is more available worldwide. The imaging agent’s cost is low, and using a ^99^Mo/^99m^Tc generator is more practical and does not require a nearby cyclotron. ^99m^Tc-EDDA/HYNIC-iPSMA (^99m^Tc-iPSMA) has been synthesized and shown to have a good affinity for tumors, especially in PCa [15,16]. There are few ^99m^Tc-iPSMA clinical studies published compared to hundreds of studies focused on PET/CT PSMA, and almost all of them were situated in the setting of biochemical relapse (BCR) [17]. Therefore, the primary aim of this study was to compare the diagnostic accuracy of ^99m^Tc-iPSMA SPECT/CT with that of ^18^F-PSMA-1007 PET/CT in patients with unfavorable intermediate-, high-, and very high-risk prostate cancer for primary diagnoses and its impact on clinical staging. The secondary objectives were to compare semiquantitative analysis (Total Tumoral Volume [TTV]) for both imaging methods and their correlation with biochemical parameters.

## 2. Materials and Methods

### 2.1. Study Design and Patients

This was a prospective controlled study where a total of eighteen (n = 18) patients with histologically confirmed prostate carcinoma, who were referred to the National Cancer Institute (INCan) between January and September 2023, were prospectively recruited. The indications for imaging were unfavorable intermediate-, high-, and very high-risk features according to current NCCN guidelines for primary staging [7]. All patients underwent ^18^F-PSMA-1007 PET/CT and ^99m^Tc-iPSMA SPECT/CT in a brief time interval (<7 days) with no particular order. Bone scans and computed tomography were not performed in our patients following NCCN prostate cancer guidelines, which indicates PSMA PET/CT can serve as an equal if not more effective alternative. Biochemical and clinical characteristics were determined in each patient and recorded. Patient exclusion criteria were as follows: (1) local or systemic prior treatment and (2) unavailable imaging data. The median age of patients was 71 (54–75) years old, and the median tPSA level was 23.3 ng/mL (4.3–920 ng/mL). Patient characteristics are summarized in Table 1. The study was approved by the Local Ethics Committee, and all participating patients signed an informed consent form.

### 2.2. Synthesis of the Tracers

^99m^Tc-pertechnetate was obtained from a ^99^Mo/^99m^Tc generator (ININ-Mexico), and radiolabeling was performed according to the standard method described by Ferro-Flores G. et al. by adding 1 mL of 02 M phosphate buffer (pH 7.0) to the lyophilized kit formulation, followed by 1 mL 740–1110 MBq of ^99m^Tc sodium pertechnetate, and incubation at 95 °C in boiling water for 10 min. ^99m^Tc-EDDA/HYNIC-iPSMA is a peptidomimetic inhibitor developed by the National Institute of Nuclear Investigations in Mexico with the following chemical structure: Lys(β-naphthyl alanine)-NH-CO-NH-Glu (Lys(Nal)-Urea-Glu), which binds to the active sites of PSMA by binding the oxygen of urea to zinc and to the carboxyl groups of the Lys(Nal)-Urea-Glu fragment to the peptides of the lateral chain of these active sites. Additionally, hydrazinonicotinamide (HYNIC) increases the lipophilicity of ^99m^Tc-EDDA/HYNIC-iPSMA, which makes it able to bind the hydrophobic sites of PSMA and with the use of N-tris[hydroxymethyl]methylglycine (tricine) and ethylenediamine acid-N,N′- diacetic (EDDA) as coligands [15].

^18^F-PSMA-1007 was synthesized as described by Cardinale J et al. [18] using the PSMA-1007 synthesis kit from ABX advanced biochemical compounds GmbH. The radiochemical purities of ^18^F-PSMA-1007 and ^99m^Tc-iPSMA were >98%, as determined by reversed-phase radio-HPLC.

### 2.3. Imaging Protocol

Whole-body PET/CT was acquired from mid-thigh to the top of the head 60 min after the injection of the radiotracer ^18^F-PSMA–1007 (370 MBq SD +/− 85 MBq) according to the clinical standard protocol. A Biograph mCT 20 Excel PET/CT scanner (Siemens Healthineers, Knoxville, TN, USA) was used. Datasets for reconstructions were: a 400 × 400 matrix (pixel size: 1.5625 × 1.5625 × 2.78 mm^3^) with Time of Flight (TOF) OSEM algorithms with 21 subsets and 4 iterations, a FWH of 6 mm, a zoom of 1.0, and a Gaussian filter with 6 mm was used. Non-enhanced CT was performed using 140 mA, 130 kV, 5 mm width, 1.5 mm pitch, reconstruction filter Br32, 2 mm slices, CARE Dose4D Care kV, and a rotation time of 0.5 s.

^99m^Tc-iPSMA SPECT/CT images were performed 5 h after radiotracer injection (740 MBq) in a SPECT/CT scanner Symbia T6 (Siemens Healthineers, Knoxville, TN, USA) using a 360-degree rotation with a non-circular orbit, step and shoot technique, 128 × 128 matrix, window of 15% centered in 140 keV with scattering correction, and 60 images of 15 s per detector. CT images were acquired from the top of the head to the middle third of the thighs. An attenuation correction map was obtained using low-dose CT parameters: a B08s AC filter for attenuation correction, rotation time of 1 s, CARE Dose4D, 80 kV reference, pitch 1.5 mm, and 5 mm slices with 2 mm reconstruction. For the display of postprocessed images, a B31s homogenous medium was used. The reconstruction of raw data was carried out using the iterative method (FLASH 3D) by the order of sets and subsets (8 iterations/4 subsets) and a Gaussian filter of 7.0.

### 2.4. Image Interpretation

The reconstructed PET/CT and SPECT/CT images were displayed on a dedicated workstation equipped with a syngo.via software VB60A (Siemens Medical Solutions, Estates, IL, USA). A maximum standardized uptake value (SUVmax) for PET/CT and target background ratio (TBR) for SPECT/CT were determined and recorded for all tracer-avid prostatic lesions. The smallest diameter (short axis) of both tracer’s avid lymph nodes was measured and recorded. Image analysis and blinded interpretation were made by two nuclear medicine physicians. The first displayed images were those of SPECT/CT and PET/CT afterward. Both reviewers used the interpretation criteria described by Rauscher [19], which considers any focal uptake of PSMA ligand higher than the surrounding background as suspicious for malignancy and not associated with physiological uptake. The same criteria were used for SPECT images. Lesion sites were divided into groups: prostate, seminal vesicle, locoregional lymph nodes, non-locoregional lymph nodes, visceral, and bone. Volumetric analysis was also performed and quantified [TTV] with lesion scout with Auto ID included within syngo.via (Siemens Healthineers, Knoxville, TN, USA), and each patient was classified according to the second version of the prostate cancer molecular imaging standardized evaluation (PROMISE V2) for the two imaging modalities [20].

### 2.5. Statistical Analysis

Results were described using median, range, and standard deviation when applicable. Student’s *t*-tests, Kruskal–Wallis tests, Spearman correlations, and Kappa agreements were performed using GraphPad Prism version 10.0.0 for Windows, GraphPad Software, Boston, MA, USA, www.graphpad.com accessed on 2 November 2023. Results with a *p*-value lower than 0.05 were considered statistically significant.

## 3. Results

### 3.1. Detection Rates

A total of 257 PSMA-expressing lesions were seen in 18 patients imaged with ^18^F-PSMA-1007 PET/CT. The location of the lesions is summarized in Table 2.

These lesions were localized in the prostate (n = 18), seminal vesicles (n = 12), locoregional lymph nodes [LR LN] (n = 62), non-locoregional [NLR] lymph nodes (n = 67), bone (n = 90), and visceral (lung and adrenal gland) (n = 8). ^99m^Tc-iPSMA detected a total of 229 (89%) lesions; both reviewers detected 100% of the lesions in the prostate (18/18), seminal vesicles (12/12), and visceral metastases (8/8); LR LN (56/62; 90%), NLR LN (57/67; 85%) and bone (78/90; 86%). The lowest diameter of positive LN detected by SPECT/CT was 3 mm (range 3–18). Table 2 and Table 3 show the performance of ^99m^Tc-iPSMA SPECT/CT in lesion detection according to sites.

### 3.2. Interobserver Agreement

An overview of the interobserver agreement for visual image interpretation is shown in Table 2. Interobserver agreement was high for the overall scan results, and the detection of primary tumors, seminal vesicles, and visceral metastases was excellent (100%). Between reviewers, there was discordance in just two LN (<4 mm) and one bone metastases (Kappa = 0.917, SE of kappa = 0.048, 95% confidence interval [CI]: From 0.823 to 1.000). Both observers identified the same regions as the location of the disease.

### 3.3. Volumetric Parameters and mi TNM classification

The Spearman-correlation coefficient between tPSA and TTV SPECT was *r* = 0.703 (*p* = 0.001), and for TTV PET it was *r* = 0.729 (*p* = 0.001). There were no statistically significant differences between TTV SPECT and TTV PET (*t* = −0.021; *p*-value = 0.49).

Molecular imaging standardized evaluation according to PROMISE V.2 was performed for the two imaging modalities. All our patients had an extraordinary agreement when classifying ^99m^Tc-iPSMA SPECT/CT versus ^18^F-PSMA PET/CT (n = 17/18; 94%). Only one patient showed a change from N1 to N2 without having a clinically significant impact. Five cases with high volume disease according to CHAARTED criteria [21] were detected by both methods (results are summarized in Figure 1 and Table 4).

### 3.4. Quantitative Analysis

To compare the contrast of primary lesions in the prostate by both methods (PET and SPECT), target-background ratio (TBR) quantifications were performed by obtaining the SUVmax of the prostate lesion divided by the SUVmax of soft tissues (gluteal region). The same procedure was performed with the counts obtained with the SPECT images. A group stratification was performed between TBR and Gleason, as well as TBR and ISUP for both methods. The values were as follows: *p =* 0.0148 and *p =* 0.053 for TBR–Gleason and *p* = 0.0148 and *p =* 0.059 for TBR-ISUP, for SPECT and PET data, respectively. The results are summarized in Figure 2.

## 4. Discussion

Emerging data is demonstrating the superiority of PSMA PET/CT over conventional imaging in prostate cancer management. PSMA imaging has shown higher efficacy in areas where morphologic imaging with CT and MRI have demonstrated deficiencies, especially in staging patients with intermediate- to high-risk prostate cancer with normal-sized lymph nodes [22]. It has also been demonstrated that ^68^Ga-PSMA has good accuracy in the diagnosis of PCa with a SUVmax cut-off of 8 in the presence of grade group >3 [23]. It is generally thought that PSMA PET/CT imaging is better than ^99m^Tc-PSMA SPECT/CT in detecting primary lesions due to its superior spatial resolution. However, in a previous study, García-Pérez et al. concluded that ^68^Ga-PSMA-11 PET/CT and ^99m^Tc-iPSMA SPECT/CT were comparable in biochemical recurrent prostate cancer patients, even in the detection of lymph nodes smaller than 10 mm [16]. Also, in a recent study performed by Iain Duncan et al., an Australian experience using ^99m^Tc-PSMA SPECT/CT in the primary diagnosis of prostate cancer and staging at biochemical recurrence after local therapy showed that using an enhanced reconstruction algorithm has a diagnostic performance similar to ^68^Ga-PSMA PET/CT and multiparametric magnetic resonance imaging in an everyday clinical setting [24]. Our study set out to determine the clinical value and accuracy of ^99m^Tc-iPSMA SPECT/CT in the primary diagnostic staging of prostate cancer. To the best of our knowledge, this is the largest prospective series of patients that have evaluated the clinical role of ^99m^Tc-iPSMA SPECT/CT in a head-to-head comparison with ^18^F-PSMA-1007 PET/CT. In this study, the primary tumor detection rate was 100% (18/18) with high interobserver agreement, showing that the performance of ^99m^Tc-iPSMA SPECT/CT is remarkable as well. These results are comparable with those published by Schmidkonz, who detected an overall detection rate of 97% PSMA-positive lesions with ^99m^Tc-MIP-1404 SPECT in 93 patients confirmed by postoperative histology [25].

However, the goal of PSMA imaging in the primary diagnostic setting is not to characterize the primary neoplasm but to detect lymph node extension and metastatic tumor burden. Caution should be taken respecting the evaluation of non-acinar histological variants (neuroendocrine, cribriform, sarcomatoid, poorly differentiated, etc.) when characterizing the primary tumor due to the variability of PSMA receptor expression [26]. Goffin et al., in a phase 2 study using ^99m^Tc-Trofolastat SPECT/CT to identify prostate cancer in intermediate- and high-risk patients undergoing extended pelvic lymph node dissection, obtained a low sensitivity (50%) and good specificity (87%) in histologically proven lymph node involvement. He also found that prostate SPECT TBR values significantly predicted lymph node involvement, with a sensitivity of 90% [27]. In our study, we demonstrated a high lesion detection rate compared with ^18^F-PSMA-1007 PET/CT, especially in locoregional lymph node involvement (51/54; 94%) and a good detection rate in non-locoregional (51/66; 77%). Those percentages correlate with a recent systematic review that evaluated PSMA-targeted surgery in PCa patients using ^99m^Tc-PSMA-I&S for detecting nodal metastases during salvage surgery; sensitivities varied widely from 50% to 100% and depended on the size and location of the lesions [28]. In our study, the lowest diameter of positive LN detected by ^99m^Tc-iPSMA SPECT/CT was 3 mm (3–18 mm), and all lymph nodes larger than 6 mm in their short axis were visualized (Table 3).

Regarding osseous involvement, in patients with intermediate- to high-risk prostate cancer, bone scans have been the most used nuclear medicine modality to seek bone metastasis. It has sufficient sensitivity; however, the specificity is low, and false positive results are common due to radiolabeled bisphosphonate uptake in a variety of benign bone lesions [29,30]. Regarding the detection of distant metastases, PSMA PET/CT has shown higher accuracy, which leads to the upstaging from a non-metastatic disease into a metastatic disease state, therefore changing the treatment strategy [31]. A recent prospective study performed by Kabunda et al. compared ^99m^Tc-PSMA scan to detect bone metastases in PCa against ^99m^Tc-MDP; they found that it was comparable and demonstrated the additional benefit of providing information on lymph nodes and visceral disease. Also, ^99m^Tc-PSMA was superior in terms of staging, biochemical relapse after radical prostatectomy, and restaging [32]. In our study, ^99m^Tc-iPSMA showed a remarkable lesion detection rate in bone and visceral metastases (n = 86/98; 87%). One patient had lung nodules and intercisural lesions < 5 mm in their longest diameter, which were accurately visualized by both methods (Figure 3). 

Another important point to highlight is the presence of unspecific bone uptake, which has been reported in almost two-thirds of patients imaged with ^18^F-PSMA-1007 PET/CT and is significantly more frequent on digital PET scanners [33]. 

Unspecific bone uptake might be misinterpreted as metastasis, with the risk of over-staging the patient, leading to inadequate therapy and overtreatment-associated costs. In our cohort group, one patient showed an unspecific bone uptake, localized in the fifth right rib, which was not visualized by ^99m^Tc-iPSMA. A CT scan control was performed after six months, with no evidence of morphological lesions and no elevation in PSA values after initial treatment (Figure 4). Therefore, it was concluded to be benign. This could infer an advantage from ^99m^Tc-iPSMA over ^18^F-PSMA-1007 PET/CT.

Interobserver applicability is an important aspect of clinical evaluation; in this study, we obtained an excellent interobserver agreement (Kappa = 0.917; 95% CI). Also, standardized evaluation for image interpretation and reporting is crucial for clinical implementation. Whereas a standardized evaluation framework has been proposed regarding PSMA PET/CT (PROMISE V2) [20], so far, there is no data regarding whole-body staging in ^99m^Tc-iPSMA SPECT/CT. Therefore, we propose using molecular imaging staging classification (mi TNM) with ^99m^Tc-iPSMA SPECT/CT in order to characterize local tumor extent, nodal and distant metastasis according to the pattern of dissemination. All of our patients had an extraordinary agreement when categorized according to this classification and also when compared with ^18^F-PSMA PET/CT (Table 4). Only one patient showed a change from N1 to N2 without having any clinical impact regarding management decisions.

The estimation of tumor volume has gained relevance in determining the response to therapy in prostate cancer due to the complexity of analyzing lesions that are not very susceptible to being studied by response criteria such as RECIST 1.1 or PCWG, in which it is only considered a numerical analysis, to determine responses. New criteria such as RECIP 1.0 already include this concept, which has proven to be an efficient alternative for this purpose [34]. In addition, its high prognostic value prior to radioligand therapy and systemic therapy with new antiandrogens or taxanes has also been demonstrated. The estimation of this functional volumetric parameter is now feasible to determine through algorithms based on artificial intelligence (AI) [35]. Our results show a comparable volumetric estimation between both methods, which strengthens its reproducibility. In addition, these findings could support the use of the TTV of ^99m^Tc-iPSMA as an imaging tool in the diagnosis, monitoring, and determination of response to therapy using AI-based auto-segmentation tools.

Our study had some limitations, such as the small number of enrolled patients and the lack of histopathological correlation in metastases. There is a need to empathize with the preliminary data analysis regarding patients with unfavorable intermediate-risk since that specific group of patients was the smallest in our study. However, our main purpose was to compare the diagnostic accuracy of ^99m^Tc-EDDA/HYNIC-iPSMA SPECT/CT related to ^18^F-PSMA-1007 PET/CT regarding detection rates. Results from subsequent studies with a larger population size will be required to support our findings.

## 5. Conclusions

^99m^Tc-iPSMA SPECT/CT is useful in the primary diagnosis of PCa. Despite its slightly lower lesion detectability compared to ^18^F-PSMA, it did not impact the clinical staging and, consequently, the initial treatment intention. Our results show that ^99m^Tc-iPSMA SPECT/CT has a remarkable interobserver agreement and no significant differences when quantifying tumor burden (TTV) compared to ^18^F-PSMA-1007 PET/CT. ^99m^Tc-iPSMA could be used as a more economical and available alternative for staging prostate cancer. Although the accessibility of PSMA PET in the world has increased, some countries with emerging economies lack sufficient infrastructure to offer this imaging modality. Our results demonstrate non-inferiority in primary staging with ^99m^Tc-iPSMA SPECT in comparison with PET in patients with prostate cancer. This could encourage its use and promote the benefit of its clinical utility.

## Figures and Tables

**Figure 1 cancers-15-05824-f001:**
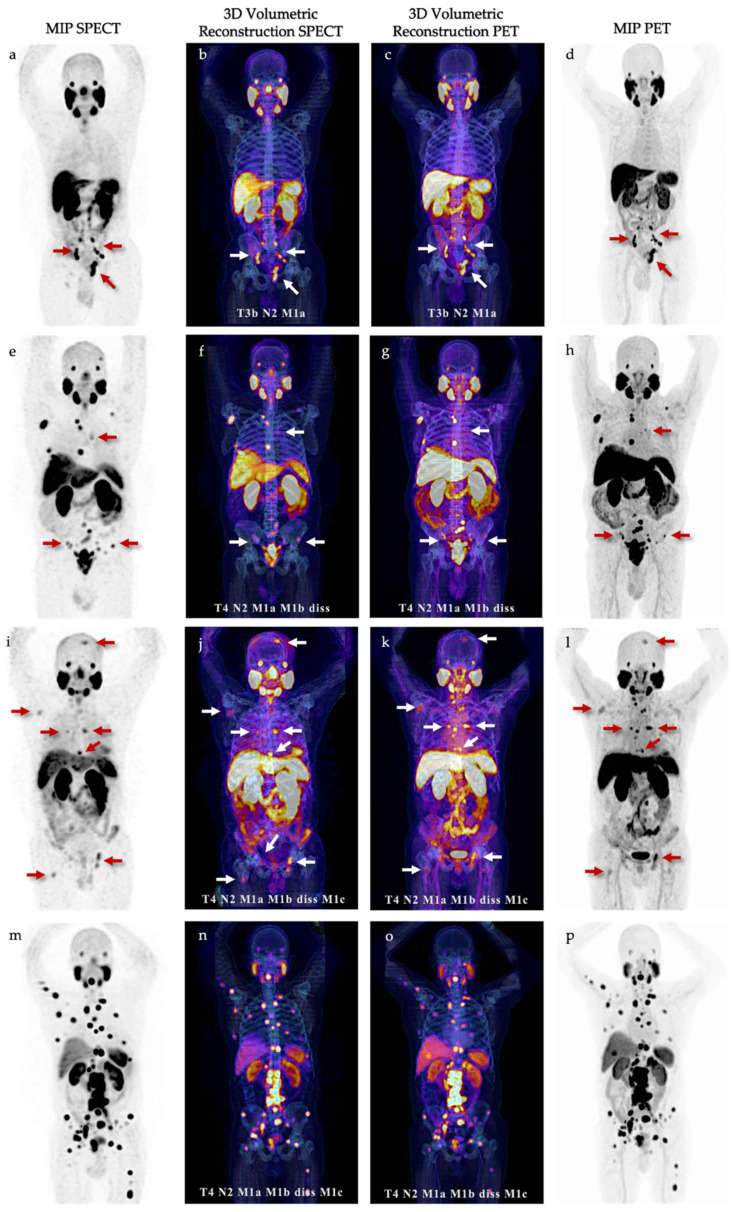
Comparative maximum intensity projection (MIP) and 3D fused volumetric reconstructions of PET/CT and SPECT/CT in most representative patients: (**a**–**d**) 74-years old, Gleason 9 (4 + 5), tPSA 25.1 ng/mL; (**e**–**h**) 71-years old, Gleason 9 (4 + 5), tPSA 21.5 ng/mL; (**i**–**l**) 75-years old, Gleason 8 (4 + 4), tPSA 189 ng/mL; (**m**–**p**) 74-years old, Gleason 9 (5 + 4), tPSA 705.7 ng/dL. Red and white arrows localize the most representative lesions.

**Figure 2 cancers-15-05824-f002:**
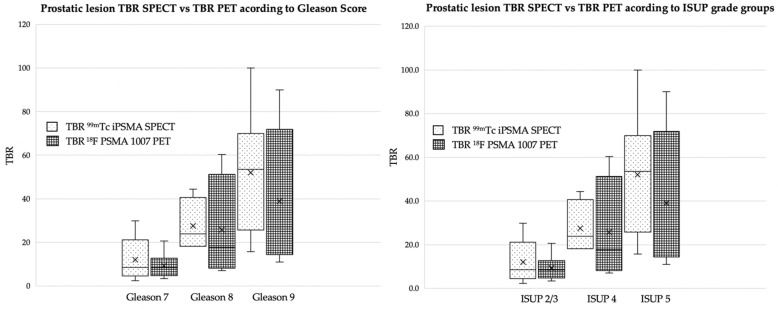
Box plot of prostatic lesions for TBR SPECT vs. TBR PET according to Gleason Score and ISUP grade groups.

**Figure 3 cancers-15-05824-f003:**
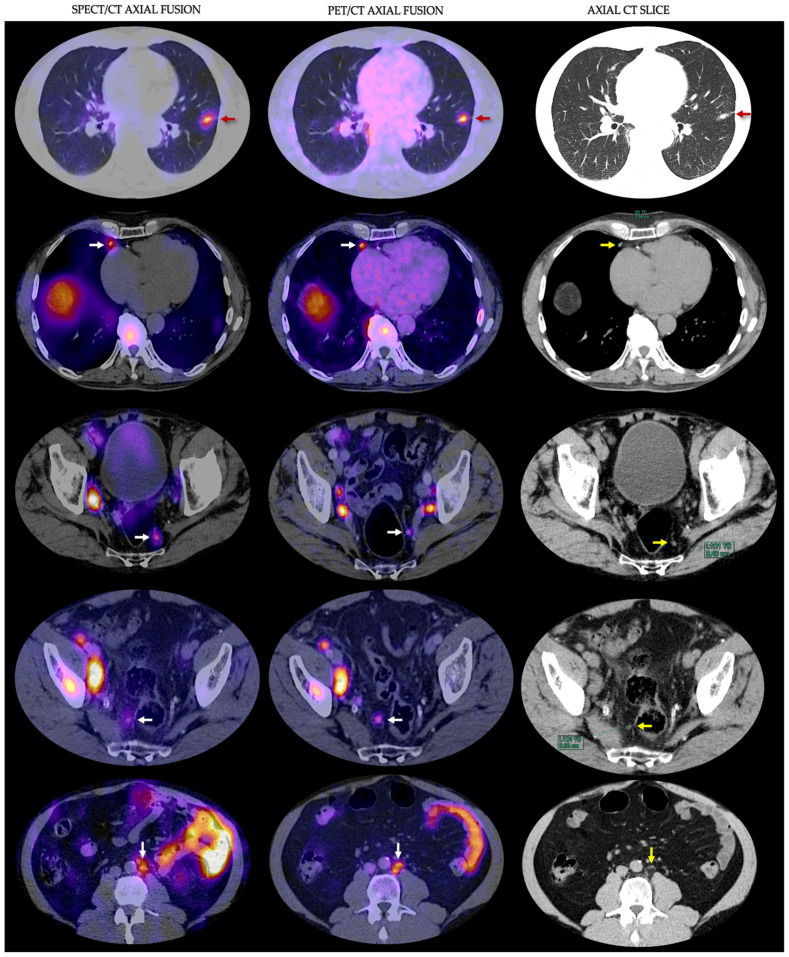
^99m^Tc-iPSMA SPECT/CT axial fusion; ^18^F-PSMA-1007 PET/CT axial fusion and axial CT slice of lowest diameter lesions detected by both methods in different patients. Red, white, and yellow arrows point to the lowest diameter lesions (<6 mm).

**Figure 4 cancers-15-05824-f004:**
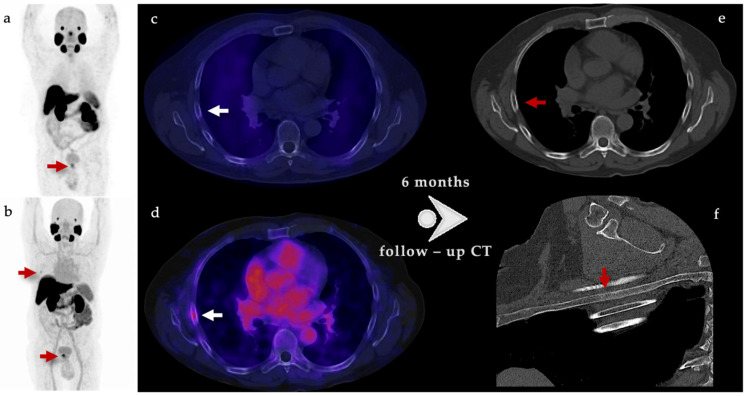
A 71-year-old male with adenocarcinoma prostate cancer Gleason 7 (4 + 3), rectal examination T2a, and tPSA 10.3 ng/mL. The risk stratification, according to NCCN, was unfavorable intermediate. (**a**) Maximum Intensity Projection (MIP) of ^99m^Tc-iPSMA shows increased uptake in prostatic lesion; (**b**) MIP of ^18^F-PSMA-1007 with increased uptake in the prostatic lesion and in right hemithorax (red arrow); (**c**) axial fusion of ^99m^Tc-iPSMA SPECT/CT with no abnormal uptake sites; (**d**) axial fusion of ^18^F-PSMA-1007 PET/CT with focal uptake in 5th right rib (white arrow), considered as unspecific bone uptake (UBU); (**e**,**f**) follow up CT scan six months later, with no evidence of blastic lesions. Therefore, the previous focal uptake was concluded to be benign.

**Table 1 cancers-15-05824-t001:** Patient’s characteristics.

Patient Characteristics	
Number of patients (n)	18
Age (years)	
Median	71 (54–75)
Mean	68.16
PSA at scan (ng/mL)	
Median	23.3 (4.3–920)
Mean	142.31
ISUP Classification	
2	n = 1 (6%)
3	n = 5 (28%)
4	n = 4 (22%)
5	n = 8 (44%)
Clinical T stage	
Tx	n = 4 (22%)
T1c	n = 2 (11%)
T2a	n = 5 (28%)
T2b	n = 1 (6%)
T2c	n = 1 (6%)
T3	n = 2 (11%)
T4	n = 3 (16%)
Risk Classification according to NCCN	
Unfavorable Intermediate	3 (17%)
High Risk	7 (39%)
Very High Risk	8 (44%)

**Table 2 cancers-15-05824-t002:** Lesions detected by ^18^F-PSMA-1007 and its correlation with lesions visualized by both observers with ^99m^Tc-iPSMA.

Site	^18^F-PSMA 1007 (n)	^99m^Tc-iPSMA (Observer 1)	^99m^Tc-iPSMA (Observer 2)
Prostate	18	18 (100%)	18 (100%)
Seminal vesicle	12	12 (100%)	12 (100%)
Locoregional lymph nodes	62	56 (90%)	55 (88%)
Non locoregional lymph nodes	67	57 (85%)	58 (86%)
Bone	90	78 (86%)	77 (85%)
Visceral	8	8 (100%)	8 (100%)
Total	257	229	228

**Table 3 cancers-15-05824-t003:** Lesions identified by PET and SPECT according to PROMISE V.2 lymph node regions.

	Region	^18^F-PSMA 1007 (n)	Lowest Diameter Detected	^99m^Tc-iPSMA (n)	Lowest Diameter Detected
mi N1/N2					
II	Internal iliac	17	2 mm	14	4 mm
EI	External iliac	22	5 mm	22	5 mm
OB	Obturator	19	3 mm	17	4 mm
PS	Presacral	1	5 mm	1	5 mm
OP	Other pelvic	3	3 mm	3	3 mm
mi M1a					
CI	Common Iliac	19	4 mm	17	5 mm
RP	Retroperitoneal	35	3 mm	28	3 mm
SD	Supradiaphragmatic	11	4 mm	10	5 mm
OE	Other extrapelvic	2	6 mm	2	6 mm

**Table 4 cancers-15-05824-t004:** Molecular imaging staging classification (mi TNM) according to PROMISE V.2 with ^18^F-PSMA-1007 PET/CT and ^99m^Tc-iPSMA SPECT/CT.

Patient Number	mi TNM PROMISE V2 PET	mi TNM PROMISE V2 SPECT
1	T3b N2 M0	T3b N1 M0
2	T3b N2 M1a M1b diss ^1^	T3b N2 M1a M1b diss ^1^
3	T3b N0 M0	T3b N0 M0
4	T3b N2 M1a	T3b N2 M1a
5	T2m N0 M0	T2m N0 M0
6	T4 N2 M1a M1b diss ^1^ M1c	T4 N2 M1a M1b diss ^1^ M1c
7	T2u N0 M0	T2u N0 M0
8	T2u N0 M1b diss ^1^	T2u N0 M1b diss ^1^
9	T2u N1 M0	T2u N1 M0
10	T4 N2 M1a M1b diss ^1^	T4 N2 M1a M1b diss ^1^
11	T3b N2 M1a	T3b N2 M1a
12	T3b N0 M0	T3b N0 M0
13	T3b N2 M1a M1b oligo ^2^	T3b N2 M1a M1b oligo ^2^
14	T2m N0 M0	T2m N0 M0
15	T2m N2 M1a	T2m N2 M1a
16	T4 N2 M1a M1b diss ^1^ M1c	T4 N2 M1a M1b diss ^1^ M1c
17	T4 N0 M0	T4 N0 M0
18	T3b N0 M0	T3b N0 M0

^1^ diss = disseminated; ^2^ oligo = oligometastatic.

## Data Availability

Data are contained within the article.

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
