# Peer review of "Diagnostic Performance of 99mTc-iPSMA SPECT/CT in the Initial Staging of Patients with Unfavorable Intermediate-, High-, and Very High-Risk Prostate Cancer: A Comparative Analysis with 18F-PSMA-1007 PET/CT"

_cancers, 2023, doi:10.3390/cancers15245824_

Round 1

Reviewer 1 Report

Comments and Suggestions for Authors

Authors should be congratulated for their work. The topic is interesting and intriguing. The manuscript is well-written and easily readable. However, minor corrections of Grammar mistakes, punctuation, and typos must be performed. The manuscript showed interesting results regarding the use of 99mTc-iPSMA SPECT/CT in the primary diagnosis of PCa patients, even if with a slightly lower detection rate compared to the gold standard imaging. This lower rate did not affect patients' staging.  The benefit of this imaging technique is represented by the lower cost, without affecting patients' staging. Therefore in the future, it will be of interest to take this diagnostic option in mind. The manuscript is suitable for publication after a minor revision.

Comments on the Quality of English Language

Minor corrections of Grammar mistakes, punctuation, and typos are needed. 

Author Response

Thanks for your comments, all Grammar mistakes, punctuation, and typos has been corrected,

Reviewer 2 Report

Comments and Suggestions for Authors

The use of novel 99mTc-labeled radiopharmaceuticals instead of PET/CT tracers can be still of value. Even if SPECT/CT is regarded as inferior to PET/CT with regard to spatial resolution, sensitivity and accuracy, 99mTc-labeled equivalents are cheaper, the labeling process is easier and, last but not least, they provide lower radiation dose to patients. Therefore, even if 68Ga- or 18F-PSMA PET/CT is quite available for the diagnosis of prostate cancer, its high cost and lower number of PET/CT centers across developing countries justifies research aimed at the evaluation of non-PET surrogate modalities.

The presented study provides data that encourage the use of 99mTc-PSMA compound for the staging of prostate cancer.  Despite the relatively low number of patients included, the paper is interesting, the study design is straight-forward. The results are clearly presented and the manuscript earned to be published. There are some issures that should be improved.

1. There is little known about the novel radiopharmaceutical and, as the compound under investigation, it should be clearly defined. How was it obtained or from whom was it purchased? Simple statement that labeling was performed according to Ferro-Flores et al is insufficient as the labeling is only the final stage of the preparation. The full name 99mTc-EDDA/HYNIC-iPSMA is present in the abstract only. Why is the ligand called iPSMA? 

2. Correlation of TBR and Gleason score and ISUP was explored but no data is presented regarding the statistical significance in the Results section and in the Fig. 2. Please provide correlation index and p values for the Gleason and ISUP subgroups.

Comments on the Quality of English Language

Minor remarks

1. List of Authors should be checked with regard to the order of names and surnames.

2. Abbreviations ISUP (line 54) and TTV (line 86) should be explained in the Introduction.

3. Line 78 imaging agent's cost is low

4. Line 136; please specify if the smallest diameter refers to the short or long axis of the lymph nodes or other lesions.

5. line 143 non-locoregional or extra-regional 

6. Line 188 between TBR adn Gleason as well as TBR and ISUP for both methods

7. Line 230 Australian

8. Line 237 "in a head-to-head comparison"

9. Line 261 It has sufficient sensitivity (not excellent)

10. Line 264 which leads to the upstaging from a non-metastatic disease...

11. It seems that abbreviations UBU and LNI do not have to be introduced into the text as they are used only few times and they are not well-known to the readers.

12. Line 314. Despite its slightly lower lesion detectability compared to F-PSMA, it did not impact the clinical staging and, consequently, the initial treatment intention.

Author Response

The presented study provides data that encourage the use of 99mTc-PSMA compound for the staging of prostate cancer.  Despite the relatively low number of patients included, the paper is interesting, the study design is straight-forward. The results are clearly presented and the manuscript earned to be published. There are some issures that should be improved.

  1. There is little known about the novel radiopharmaceutical and, as the compound under investigation, it should be clearly defined. How was it obtained or from whom was it purchased?

99mTc-EDDA/HYNIC-iPSMA is a peptidomimetic inhibitor developed by the National Institute of Nuclear Investigations in Mexico, Actually this molecule is available in Mexico for clinical use and is our formal provider.

  1. Simple statement that labeling was performed according to Ferro-Flores et al is insufficient as the labeling is only the final stage of the preparation.

The full name 99mTc-EDDA/HYNIC-iPSMA is present in the abstract only. Why is the ligand called iPSMA?.  Lys(β-naphthyl alanine)-NH-CO-NH-Glu (Lys(Nal)-Urea-Glu)  or 99mTc-EDDA/HYNIC-iPSMA  binds to the active sites of PSMA by binding the oxygen of urea to zinc and to the carboxyl groups of the Lys(Nal)-Urea-Glu fragment to the peptides of the lateral chain of these active sites, additionally , hydrazinonicotinamide (HYNIC) increases the lipophilicity of 99mTc-EDDA/HYNIC-iPSMA, which makes it able to bind the hydrophobic sites of PSMA and with the use of N-tris[hydroxymethyl]methylglycine (tricine) and ethylenediamine acid-N,N ́- diacetic (EDDA) as coligands. ININ the owner of patent has proposed the short name iPSMA for their molecule.

  1. Correlation of TBR and Gleason score and ISUP was explored but no data is presented regarding the statistical significance in the Results section and in the Fig. 2. Please provide correlation index and p values for the Gleason and ISUP subgroups.

To evaluate differences between Gleason score and ISUP groups and iPSMA/PSMA1007 TBR´s que use Kruskal–Wallis test, p values are now consigned in paper results.

Minor results has been corrected

Reviewer 3 Report

Comments and Suggestions for Authors

The manuscript is interesting but some points should be improved

1. The accuracy of 99mTc-iPSMA SPECT/CT99mTc and 18F PSMA PET/CT in the diagnosis of prostate cancer can not be evaluated because 83% of the patients evaluated had an high/very high risk PCa; the Authors could compare the accuracy only in PCa staging 

2. Data about conventional imaging staging (CT and bone scan) have not been reported 

3. The SUVmax cut-off indicated in the literature for the diagnosis of PCa should be reported ( Pepe Pet al: F. 68Ga-PSMA PET/CT and Prostate Cancer Diagnosis: Which SUVmax Value? In Vivo. 2023 May-Jun;37(3):1318-1322. doi: 10.21873/invivo.13211. PMID: 37103095; PMCID: PMC10188025 - Pepe Pet al Targeted Biopsy in Men High Risk for Prostate Cancer: 68Ga-PSMA PET/CT Versus mpMRI. Clin Genitourin Cancer. 2023 Jun 19:S1558-7673(23)00146-5. doi: 10.1016/j.clgc.2023.06.007. Epub ahead of print. PMID: 37394379)

3. The number of patients evaluated is very limited (as reported) but it should be underlined (preliminary data analysis) hoping to evaluate the results in men with intermediate risk in the intere prostate specimen 

4. The low accuracy of PSMA PET/CT in the diagnosis and staging in men with non acinar high hisk PCa (ductal prostatic adenocarcinoma, cribiform PCa) should be reported in the Discussion

Author Response

  1. The accuracy of 99mTc-iPSMA SPECT/CT99mTc and 18F PSMA PET/CT in the diagnosis of prostate cancer can not be evaluated because 83% of the patients evaluated had an high/very high risk PCa; the Authors could compare the accuracy only in PCa staging.

We have corrected the groups included in paper, considering unfavorable intermediate, high and very high risk, the aim of our work is to compare rate of detection in comparison with PSMA-1007, considered as a reference modality for staging prostate cancer in those groups and according to guidelines.

  1. Data about conventional imaging staging (CT and bone scan) have not been reported,

CT and bone scan was not performed according to recommendation consigned in PROS-10 of NCCN guidelines version 4.2023 that establish “PSMA PET/CT or PSMA/MRI are preferred for bone and soft tissue (full body) imaging.” this was included in line 91 (material and methods) .

  1. The SUVmax cut-off indicated in the literature for the diagnosis of PCa should be reported ( Pepe Pet al: F. 68Ga-PSMA PET/CT and Prostate Cancer Diagnosis: Which SUVmax Value? In Vivo. 2023 May-Jun;37(3):1318-1322. doi: 10.21873/invivo.13211. PMID: 37103095; PMCID: PMC10188025 - Pepe Pet al Targeted Biopsy in Men High Risk for Prostate Cancer: 68Ga-PSMA PET/CT Versus mpMRI. Clin Genitourin Cancer. 2023 Jun 19:S1558-7673(23)00146-5. doi: 10.1016/j.clgc.2023.06.007. Epub ahead of print. PMID: 37394379)

Data about this was included in line 237

  1. The number of patients evaluated is very limited (as reported) but it should be underlined (preliminary data analysis) hoping to evaluate the results in men with intermediate risk in the intere prostate specimen.

This point now has been underlined in discussion. Highlighting the need of develop a investigation in this sub group.

  1. The low accuracy of PSMA PET/CT in the diagnosis and staging in men with non acinar high hisk PCa (ductal prostatic adenocarcinoma, cribiform PCa) should be reported in the Discussion:

We omit to consign this information and in this new version we add the next paragraph acording to your suggest: “Caution should be taken respecting the evaluation of non-acinar histological variants (neuroendocrine, cribiform, sarcomatoid, poorly differentiated, etc…) when characterizing the primary tumor, due to the variability of PSMA receptors expression.  (line 259)

Round 2

Reviewer 3 Report

Comments and Suggestions for Authors

The manuscript has been improved